# `MonarchAttention`: Zero-Shot Conversion to Fast, Hardware-Aware Structured Attention

**Can Yaras**
University of Michigan
cjyaras@umich.edu

**Alec S. Xu**
University of Michigan
alecx@umich.edu

**Pierre Abillama**
University of Michigan
pabillam@umich.edu

**Changwoo Lee**
University of Michigan
cwoolee@umich.edu

**Laura Balzano**
University of Michigan
girasole@umich.edu

## Abstract

Transformers have achieved state-of-the-art performance across various tasks, but suffer from a notable quadratic complexity in sequence length due to the attention mechanism. In this work, we propose `MonarchAttention` – a novel approach to sub-quadratic attention approximation via Monarch matrices, an expressive class of structured matrices. Based on the variational form of softmax, we describe an efficient optimization-based algorithm to compute an approximate projection of softmax attention onto the class of Monarch matrices with $\Theta(N\sqrt{N}d)$ computational complexity and $\Theta(Nd)$ memory/IO complexity. Unlike previous approaches, `MonarchAttention` is both (1) transferable, yielding minimal performance loss with no additional training, even when replacing every attention layer of the Transformer, and (2) hardware-efficient, utilizing the highest-throughput tensor core units on modern GPUs. With optimized kernels, `MonarchAttention` achieves substantial speed-ups in wall-time over `FlashAttention-2`: $1.4\times$ for shorter sequences ($N = 256$), $4.5\times$ for medium-length sequences ($N = 4K$), and $8.2\times$ for longer sequences ($N = 16K$). We demonstrate the quality of `MonarchAttention` on diverse tasks and architectures in vision and language problems, showing that it flexibly and accurately approximates softmax attention in a variety of contexts. Our code is available at `https://github.com/cjyaras/monarch-attention`.

## 1 Introduction

Over the past decade, Transformers (Vaswani et al., 2017) have become the dominant architecture for generating and processing various data modalities, such as text (Brown et al., 2020), images (Dosovitskiy et al., 2021), and speech (Radford et al., 2023). Central to the Transformer's success is *attention*, the mechanism through which complex interactions within sequential data are captured through weighted combinations of embeddings at every position in the sequence. Famously, the attention mechanism has a quadratic-time complexity $\Theta(N^2d)$ in the length of the sequence $N$, where $d$ is the head dimension, which is a key bottleneck for both training and inference, particularly in long sequence problems. To address this, numerous works have proposed sub-quadratic substitutes for attention. Yet, such approaches are either (1) not transferable, requiring training from scratch or fine-tuning of existing models, or (2) do not yield speed-ups in practice (except on extremely long sequences) due to a gap between theoretical complexity and practical considerations for modern GPUs, especially compared to highly optimized implementations (Dao et al., 2022b).

In this work, we propose `MonarchAttention`: a novel sub-quadratic attention substitute

39th Conference on Neural Information Processing Systems (NeurIPS 2025).

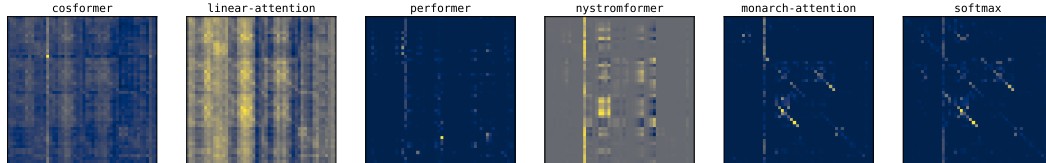

Figure 1: **Approximation of softmax attention via** `MonarchAttention`**.** By directly optimizing the softmax variational objective constrained to Monarch matrices, `MonarchAttention` yields accurate zero-shot approximation to softmax attention compared to other hardware-friendly, efficient attention baselines. Attention maps extracted from RoBERTa on the SQuAD dataset in Section 4.

based on approximating the attention matrix via *Monarch* matrices (Dao et al., 2022a), a class of expressive structured matrices. At first glance, this is computationally infeasible – for sequence length $N$, computing an exact projection onto the set of Monarch matrices has a super-quadratic $O(N^2\sqrt{N})$-time complexity, not to mention that we need to form the entire $N \times N$ attention matrix. Instead, we reframe the computation of the attention matrix as an optimization problem in terms of the variational form of softmax, and exploit low-dimensional structure in the variational objective when constrained to the set of Monarch matrices – this yields a sub-quadratic $\Theta(N\sqrt{N}d)$-time approximation, where $d$ is the head dimension. This approach combines two optimization-based improvements for its success: the variational form of the softmax nonlinearity with a structured factorization approach analogous to those for low-rank approximation of a matrix (Chi et al., 2019), where rather than computing a full SVD and truncating to the desired rank, one can more efficiently minimize a Frobenius norm objective constrained to the set of low-rank matrices. We briefly review prior work on structured matrices, including Monarch matrices, as well as existing approaches to efficient attention.

**Structured & Monarch Matrices.** We use the phrase "structured matrices" to mean those that admit sub-quadratic storage and matrix-vector multiplication, such as low-rank or sparse matrices. There are many useful classes of structured matrices, such as those with *low displacement rank* (Kailath et al., 1979), which includes Toeplitz, Hankel, Vandermonde, Cauchy matrices (Pan, 2001); *orthogonal polynomial transforms* (Chihara, 2014), which includes discrete Fourier/cosine and Hadamard transforms; *butterfly factorizations* (Dao et al., 2019), which implement fast matrix-vector multiplication via a recursive divide-and-conquer algorithm similar to that of fast Fourier transforms (FFTs); and *Monarch matrices*, an expressive family of structured matrices (generalizing butterfly matrices and thereby many fast transforms) that overcome unfavorable memory access patterns typical to FFT-like algorithms by implementing matrix products via batched dense matrix multiplications (also called matmuls) on fast tensor cores found in modern GPUs.

**Sub-Quadratic Attention.** Nearly all approaches to sub-quadratic attention approximate the attention matrix by a structured matrix, specifically low-rank and/or sparse.

- **Low-Rank.** Motivated by Johnson-Lindenstrauss embeddings, Wang et al. (2020) propose sketching the key and value matrices along the sequence dimension via learnable projections. Katharopoulos et al. (2020) introduce *linear attention*, where the exponential kernel is approximated via inner products of queries and keys lifted via some feature map. Several follow-up works proposed various feature maps, such as the exponential linear unit (ELU) (Katharopoulos et al., 2020), random positive features (Choromanski et al., 2021), rectified linear unit (ReLU) with cosine reweighting (Qin et al., 2022), and learnable single-layer multi-layer perceptrons (MLPs) (Zhang et al., 2024). Xiong et al. (2021) use the Nyström method for computing low-rank approximations by sampling rows and columns.

- **Sparse.** Child et al. (2019) introduce sparsity by applying fixed, structured sparse masks on the attention matrix. In particular, Chen et al. (2022) propose a particular *block* butterfly matrix for the sparse mask, which is more hardware-friendly at the cost of reduced expressiveness. Those that do not enforce a structure on the sparsity pattern include Kitaev et al. (2020); Daras et al. (2020) where they utilize locality-sensitive hashing (LSH) on shared query/key vectors to only compute attention within clusters of similar tokens.

- **Low-Rank + Sparse.** Inspired by robust PCA, Chen et al. (2021) decompose the attention matrix into a sum of two matrices: an unstructured sparse component using LSH and a low-rank component that is constructed via linear attention. Han et al. (2024) propose to subsample columns of the non-normalized attention matrix based on row norms of the value matrix, while estimating the softmax normalization factors from a few large elements via LSH.

We note that there are significant drawbacks to the approaches described above. Pure low-rank methods are often fast and hardware-friendly, but are typically not suitable as drop-in replacements for attention in pre-trained Transformers due to the prevalence of "strongly diagonal", high-rank attention matrices where attention weights are concentrated locally in a sequence. Making up for this with a fixed sparsity pattern does not allow for data-dependent support of the attention matrix, necessary for zero-shot conversion. Finally, sparsity/LSH-based approaches that do not have a fixed sparsity pattern improve on accuracy over low-rank approximations but suffer from significant overhead due to GPU incompatibility. `MonarchAttention`, on the other hand, achieves the best of both worlds: it is fast and hardware-friendly due to utilization of tensor cores for batched matmuls, while computing highly accurate approximations to the extent that it can directly replace softmax attention with no additional training.

We conclude this section by discussing closely related works. Dao et al. (2022b) propose `FlashAttention`, an IO-aware streaming algorithm for computing exact softmax attention. We show in Section 3 that each step of `MonarchAttention` can be written as a `FlashAttention`-like computation, allowing for similar IO savings to `FlashAttention` – in fact, we demonstrate that `MonarchAttention` achieves a strictly better worst-case IO complexity compared to `FlashAttention`. We also note that Dao et al. (2022b) propose to further accelerate `FlashAttention` using block butterfly attention masks, so `MonarchAttention` can be viewed as a generalization of block-sparse `FlashAttention` to more general Monarch matrices. Finally, `MonarchAttention` is closely related to Monarch Mixer (Fu et al., 2023), a mixer-type architecture (Tolstikhin et al., 2021) that utilizes Monarch instead of dense matrices for token and channel mixing. `MonarchAttention` also uses Monarch matrices for mixing tokens – however, it is based on the attention operation which is *data-dependent*, unlike Monarch Mixer.

**Organization.** In Section 2, we discuss preliminaries on (softmax) attention, and Monarch matrices. In Section 3, we describe the `MonarchAttention` algorithm and implementation. In Section 4, we evaluate `MonarchAttention` in a variety of settings for zero-shot conversion to sub-quadratic attention and benchmark its implementation. In Section 5, we discuss limitations and future directions.

## 2 Preliminaries

**Notation.** We use $[N]$ to denote the index set $\{1, 2, \ldots, N\}$. We use $\Delta^N$ to denote the $(N-1)$ dimensional unit simplex, given by $\Delta^N = \{\boldsymbol{a} \in \mathbb{R}^N : \boldsymbol{a} \succeq 0, \langle \mathbf{1}_N, \boldsymbol{a} \rangle = 1\}$. We denote the $m \times m$ identity matrix by $\boldsymbol{I}_m$. We use the notation $\boldsymbol{A}_{ijk}$ to denote an element of a 3-way tensor, and $\boldsymbol{A}_{i,:,k}$ to denote a slice. We use $\delta_{kl}$ to denote the Kronecker delta that is 1 if $k = l$ and otherwise 0.

**Softmax.** The softmax function $\mathbb{R}^N \to \Delta^N$ maps $N$ real numbers to the $(N-1)$-dimensional unit simplex, and is defined as

$$[\mathrm{softmax}(\boldsymbol{z})]_i := \frac{\exp(\boldsymbol{z}_i)}{\sum_j \exp(\boldsymbol{z}_j)}, \ \forall i \in [N]. \tag{1}$$

An alternative definition (Blondel et al., 2019) is given by the following variational form:

$$\mathrm{softmax}(\boldsymbol{z}) := \arg\max_{\boldsymbol{a} \in \Delta^N} \langle \boldsymbol{a}, \boldsymbol{z} \rangle + H(\boldsymbol{a}), \tag{2}$$

where $H(\boldsymbol{a}) = -\sum_i \boldsymbol{a}_i \log \boldsymbol{a}_i$ is Shannon entropy. See Appendix A for equivalence of (1) and (2).

**Attention.** Given query, key, value matrices $\boldsymbol{Q}, \boldsymbol{K}, \boldsymbol{V} \in \mathbb{R}^{N \times d}$, where $N$ is the sequence length and $d$ is the head dimension, a single head of standard softmax attention[1] computes

$$\boldsymbol{O} = \mathrm{softmax}\left(\boldsymbol{Q}\boldsymbol{K}^\top\right)\boldsymbol{V}, \tag{3}$$

---

[1]Typically, the $\boldsymbol{Q}\boldsymbol{K}^\top$ matrix is scaled by a factor of $d^{-1/2}$, but this can be absorbed into $\boldsymbol{Q}$.

where the softmax function is applied across rows. The computational complexity of attention is $\Theta(N^2 d)$ for each forward pass, because the matrices $\boldsymbol{Q}, \boldsymbol{K}, \boldsymbol{V}$ are data-dependent.

**Monarch Matrices.** Given $N = m \times b$ for integers $m, b$, we define a block rank-one matrix $\boldsymbol{B} \in \mathbb{R}^{N \times N}$ as

$$\boldsymbol{B} = \begin{bmatrix} \boldsymbol{B}_{11} & \dots & \boldsymbol{B}_{1m} \\ \vdots & \ddots & \vdots \\ \boldsymbol{B}_{b1} & \dots & \boldsymbol{B}_{bm} \end{bmatrix}, \quad \text{where} \quad \boldsymbol{B}_{jk} = \boldsymbol{L}_{jk} \boldsymbol{R}_{kj}^\top \in \mathbb{R}^{m \times b}$$

for some $\boldsymbol{L}_{jk} \in \mathbb{R}^m, \boldsymbol{R}_{kj} \in \mathbb{R}^b$ for $j \in [b]$ and $k \in [m]$. It follows that

$$\boldsymbol{B} = \begin{bmatrix} \boldsymbol{L}_1^\top & & & \\ & \boldsymbol{L}_2^\top & & \\ & & \ddots & \\ & & & \boldsymbol{L}_b^\top \end{bmatrix} \boldsymbol{P} \begin{bmatrix} \boldsymbol{R}_1 & & & \\ & \boldsymbol{R}_2 & & \\ & & \ddots & \\ & & & \boldsymbol{R}_m \end{bmatrix},$$

where $\boldsymbol{L}_j \in \mathbb{R}^{m \times m}$ for $j \in [b]$ and $\boldsymbol{R}_k \in \mathbb{R}^{b \times b}$ for $k \in [m]$, and $\boldsymbol{P} \in \mathbb{R}^{N \times N}$ is a "transpose"[2] permutation matrix whose $(i+1)$-th row is given by $\boldsymbol{e}_{\sigma(i)+1}$ where

$$\sigma(i) = b \cdot (i \bmod m) + \left\lfloor \frac{i}{m} \right\rfloor, \quad i \in \{0, \dots, N-1\}.$$

Given the above, a Monarch matrix $\boldsymbol{M} \in \mathbb{R}^{N \times N}$ is given by $\boldsymbol{M} = \boldsymbol{P}^\top \boldsymbol{B}$ – in other words, it is a row-permuted block rank-one matrix. When $m = b = \sqrt{N}$, storing such a matrix requires only $\Theta(N\sqrt{N})$ space, while matrix multiplication (matmul) with a matrix $\boldsymbol{V} \in \mathbb{R}^{N \times d}$ can be computed efficiently in $\Theta(N\sqrt{N}d)$ operations (as opposed to $\Theta(N^2 d)$ for dense matrices) with batched matmuls and transposes:

$$\boldsymbol{M}\boldsymbol{V} = \boldsymbol{O}, \quad \text{where} \quad \boldsymbol{O}_{b \cdot (l-1)+j, v} = \sum_k \boldsymbol{L}_{jkl} \boldsymbol{Y}_{jkv}, \quad \boldsymbol{Y}_{jkv} = \sum_i \boldsymbol{R}_{kji} \boldsymbol{V}_{b \cdot (k-1)+i, v} \tag{4}$$

for $i, j \in [b], k, l \in [m]$. A useful characterization of $\boldsymbol{M}$ is in block form:

$$\boldsymbol{M} = \begin{bmatrix} \boldsymbol{M}_{11} & \dots & \boldsymbol{M}_{1m} \\ \vdots & \ddots & \vdots \\ \boldsymbol{M}_{m1} & \dots & \boldsymbol{M}_{mm} \end{bmatrix}, \tag{5}$$

where $\boldsymbol{M}_{lk} \in \mathbb{R}^{b \times b}$, $[\boldsymbol{M}_{lk}]_{ji} = \boldsymbol{L}_{jkl} \boldsymbol{R}_{kji}, \forall i, j \in [b], k, l \in [m]$.

## 3 MonarchAttention

The main goal of `MonarchAttention` is to find a Monarch matrix $\boldsymbol{M} \in \mathbb{R}^{N \times N}$ in $o(N^2 d)$ time such that $\boldsymbol{M} \approx \text{softmax}(\boldsymbol{Q}\boldsymbol{K}^\top)$. Then, we can approximately compute the output $\boldsymbol{O} = \boldsymbol{M}\boldsymbol{V}$ using efficient matmul. We can do this by viewing the softmax operation as an optimization problem via its variational form (2), whose objective can be efficiently maximized with exact alternating steps when constrained to Monarch matrices. As shown in Figure 1, this yields highly accurate approximations to the softmax attention matrix.

**Softmax Objective.** First, from (2) we can write

$$\sigma(\boldsymbol{Q}\boldsymbol{K}^\top) = \underset{\boldsymbol{A} \in \Delta^{N \times N}}{\arg\max} \ f(\boldsymbol{A}; \boldsymbol{Q}, \boldsymbol{K}) := \langle \boldsymbol{A}, \boldsymbol{Q}\boldsymbol{K}^\top \rangle + H(\boldsymbol{A}), \tag{6}$$

where $\Delta^{N \times N}$ denotes a matrix whose rows lie on $\Delta^N$, and $H(\boldsymbol{A}) = -\sum_{i,j} \boldsymbol{A}_{ij} \log \boldsymbol{A}_{ij}$. Here we demonstrate how to handle this objective function when $\boldsymbol{A}$ has Monarch structure, and defer discussion of the simplex constraint to the next paragraph. For a dense matrix $\boldsymbol{A}$, computing $f(\boldsymbol{A}; \boldsymbol{Q}, \boldsymbol{K})$

---

[2]$\boldsymbol{P}\boldsymbol{x}$ corresponds to row-major reshaping $\boldsymbol{x} \in \mathbb{R}^N$ to $\mathbb{R}^{m \times b}$, transposing to $\mathbb{R}^{b \times m}$, then row-major flattening back to $\mathbb{R}^N$. See Appendix B for an example.

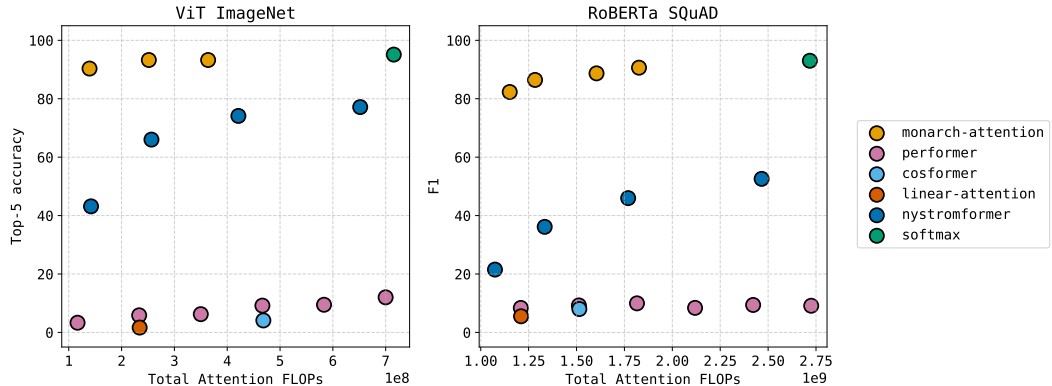

Figure 2: **Zero-shot conversion of attention layers for image classification and question answering.** We vary hyperparameters for various baselines to evaluate model quality vs compute tradeoff. *Left.* Top-5 accuracy vs. total attention FLOPs across all layers for ViT on ImageNet. *Right.* F1 score vs total attention FLOPs across all layers for RoBERTa on SQuAD.

requires $\Theta(N^2 d)$ operations, which is the same as computing $\sigma(\boldsymbol{Q}\boldsymbol{K}^\top)$ directly. However, when $\boldsymbol{A}$ is a Monarch matrix $\boldsymbol{M} = \boldsymbol{P}^\top \boldsymbol{B}$, we have

$$f(\boldsymbol{P}^\top \boldsymbol{B}; \boldsymbol{Q}, \boldsymbol{K}) = \langle \boldsymbol{P}^\top \boldsymbol{B}, \boldsymbol{Q}\boldsymbol{K}^\top \rangle + H(\boldsymbol{P}^\top \boldsymbol{B})$$
$$= \langle \boldsymbol{B}, \widetilde{\boldsymbol{Q}}\boldsymbol{K}^\top \rangle + H(\boldsymbol{B}) = \sum_{j,k} f(\boldsymbol{B}_{jk}; \widetilde{\boldsymbol{Q}}_j, \boldsymbol{K}_k),$$

where $\widetilde{\boldsymbol{Q}} = \boldsymbol{P}\boldsymbol{Q}$, and $\widetilde{\boldsymbol{Q}}_j \in \mathbb{R}^{m \times d}, \boldsymbol{K}_k \in \mathbb{R}^{b \times d}$ are the $j$-th and $k$-th block of rows of $\widetilde{\boldsymbol{Q}}, \boldsymbol{K}$ respectively. Then, for each $j \in [b], k \in [m]$ we evaluate $f$ on the rank-one matrix $\boldsymbol{B}_{jk} = \boldsymbol{L}_{jk}\boldsymbol{R}_{kj}^\top$:

$$f(\boldsymbol{B}_{jk}; \widetilde{\boldsymbol{Q}}_j, \boldsymbol{K}_k) = \langle \boldsymbol{L}_{jk}\boldsymbol{R}_{kj}^\top, \widetilde{\boldsymbol{Q}}_j\boldsymbol{K}_k^\top \rangle - \sum_{l,i} \boldsymbol{L}_{jkl}\boldsymbol{R}_{kji} \log(\boldsymbol{L}_{jkl}\boldsymbol{R}_{kji})$$
$$= \langle \widetilde{\boldsymbol{Q}}_j^\top \boldsymbol{L}_{jk}, \boldsymbol{K}_k^\top \boldsymbol{R}_{kj} \rangle - \sum_{l,i} \boldsymbol{L}_{jkl}\boldsymbol{R}_{kji} \log \boldsymbol{L}_{jkl} - \sum_{l,i} \boldsymbol{L}_{jkl}\boldsymbol{R}_{kji} \log \boldsymbol{R}_{kji}$$
$$= \langle \widetilde{\boldsymbol{Q}}_j^\top \boldsymbol{L}_{jk}, \boldsymbol{K}_k^\top \boldsymbol{R}_{kj} \rangle + \left(\mathbf{1}^\top \boldsymbol{R}_{kj}\right) \cdot H(\boldsymbol{L}_{jk}) + \left(\mathbf{1}^\top \boldsymbol{L}_{jk}\right) \cdot H(\boldsymbol{R}_{kj}).$$

Thus, for each $j \in [b], k \in [m]$ we only need $\Theta((m+b)d)$ operations to compute $f(\boldsymbol{B}_{jk}; \widetilde{\boldsymbol{Q}}_j, \boldsymbol{K}_k)$ due to $\widetilde{\boldsymbol{Q}}_j^\top \boldsymbol{L}_{jk}$ and $\boldsymbol{K}_k^\top \boldsymbol{R}_{kj}$. We emphasize that the rank-one structure implies separability of the entropy term, meaning we can compute the entropy on $\boldsymbol{L}_{jk}$ and $\boldsymbol{R}_{kj}$ individually and avoid the need to materialize $\boldsymbol{B}_{jk}$, which would incur $\Theta(mb)$ cost as opposed to $\Theta(m+b)$. Since there are $m \cdot b$ many $\boldsymbol{B}_{jk}$ matrices, we have in total $\Theta((m^2 b + b^2 m)d)$ operations to compute $f(\boldsymbol{M}; \boldsymbol{Q}, \boldsymbol{K})$, which for $m = b = \sqrt{N}$ is $\Theta(N\sqrt{N}d)$, improving on the dense computation by a factor of $\sqrt{N}$.

**Alternating Maximization with Constraints.** We will now explain the alternating maximization approach for optimizing $f$. When $\boldsymbol{L}$ is fixed, the objective is concave in $\boldsymbol{R}$, and vice-versa – therefore, we can derive closed form expressions via KKT conditions for $\boldsymbol{L}$ and $\boldsymbol{R}$ that maximize $f$ with one of $\boldsymbol{L}$ or $\boldsymbol{R}$ fixed, which will constitute a single update step. Evaluating (and therefore differentiating) $f$ w.r.t. $\boldsymbol{L}$ and $\boldsymbol{R}$ can be done in $\Theta(N\sqrt{N}d)$ time, which will be the same complexity as one of these steps. For $T$ steps, this will require $\Theta(TN\sqrt{N}d)$ computation; provided that $T = o(\sqrt{N})$, this will still be sub-quadratic. However, the constraint $\boldsymbol{M} \in \Delta^{N \times N}$ presents a challenge in its current form, since this requires materializing $\boldsymbol{M}$ to check that each entry is non-negative. Instead, we use the fact that

$$\boldsymbol{L}_{j,:,l} \in \Delta^m, \ \boldsymbol{R}_{kj} \in \Delta^b, \ \forall j \in [b], \forall k, l \in [m] \implies \boldsymbol{M} \in \Delta^{N \times N},$$

i.e., slices of $\boldsymbol{L}, \boldsymbol{R}$ individually lying on the unit simplex is sufficient to enforce the constraint on $\boldsymbol{M}$. This is easily seen from (5) – obviously if $\boldsymbol{L}_{jkl}, \boldsymbol{R}_{kji} \geq 0$, then $[\boldsymbol{M}_{lk}]_{ji} \geq 0$. Moreover, this also

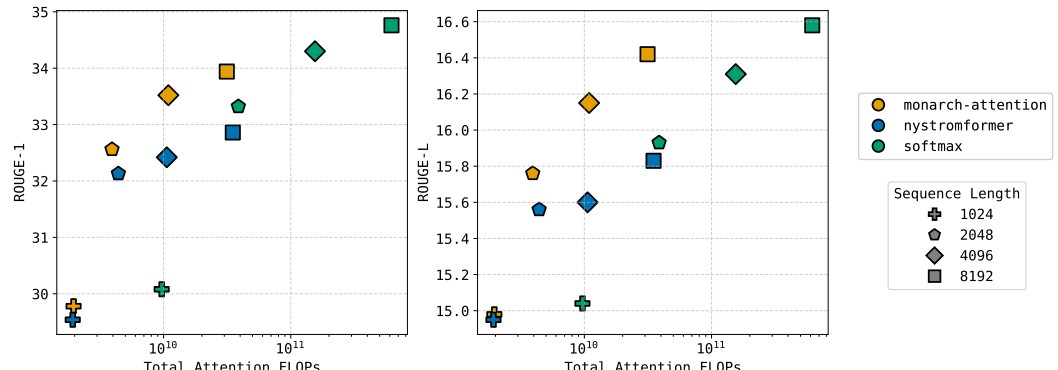

Figure 3: **Zero-shot conversion of attention layers for long sequence summarization.** We vary the sequence length of the text to be summarized to evaluate model quality vs compute tradeoff. We report recall-based ROUGE-1 and ROUGE-L scores vs. total attention FLOPs across all layers for BART on BookSum-chapters.

enforces the sum-to-one constraint, as rows of $M$ sum as

$$\sum_{k,i}[M_{lk}]_{ji} = \left(\sum_k L_{jkl}\right)\left(\sum_i R_{kji}\right) = 1.$$

We now present the updates for $L, R$. Initializing $L_{jkl}^{(0)} = \delta_{kl}$ as identity, we have

$$R^{(t)} = \text{softmax}_i(Z_R^{(t)}), \quad Z_{R,kji}^{(t)} = \beta_{R,kji}^{(t)}/c_{R,kj}^{(t)}, \tag{7}$$

$$L^{(t)} = \text{softmax}_k(Z_L^{(t)}), \quad Z_{L,jkl}^{(t)} = \beta_{L,jkl}^{(t)} - c_{L,jk}^{(t)}, \tag{8}$$

for $t \in [T]$, where $\text{softmax}_k, \text{softmax}_i$ are applied along $k$ and $i$ index dimensions respectively, and

$$\beta_{R,kji}^{(t)} = \sum_v \alpha_{R,kjv}^{(t)} \overline{K}_{kiv}, \quad \alpha_{R,kjv}^{(t)} = \sum_l L_{jkl}^{(t-1)} \overline{Q}_{jlv}, \quad c_{R,kj}^{(t)} = \sum_l L_{jkl}^{(t-1)}, \tag{9}$$

$$\beta_{L,jkl}^{(t)} = \sum_v \alpha_{L,jkv}^{(t)} \overline{Q}_{jlv}, \quad \alpha_{L,jkv}^{(t)} = \sum_i R_{kji}^{(t)} \overline{K}_{kiv}, \quad c_{L,jk}^{(t)} = \sum_i R_{kji}^{(t)} \log R_{kji}^{(t)}, \tag{10}$$

where $\overline{Q}_{jl}, \overline{K}_{ki} \in \mathbb{R}^d$ are the $(b \cdot (l-1) + j)$-th and $(b \cdot (k-1) + i)$-th rows of $Q$ and $K$ respectively. The full derivation is provided in Appendix C.1. After $T$ steps, we obtain the final Monarch approximation $M^{(T)} \approx \sigma(QK^\top)$ with factors $L^{(T)}$ and $R^{(T)}$, from which we output $M^{(T)}V$ using (4). A naïve implementation of the full algorithm is provided in Appendix C.2. We discuss in Appendix C.3 how padding can be incorporated into `MonarchAttention` for when $N$ is not divisible by $b$. We empirically demonstrate the fast convergence of `MonarchAttention` in Appendix E.1.

**Implementation.** To minimize data movement and memory usage on GPU, we do not materialize $L$ or $R$ in high-bandwidth memory (HBM). In addition to $Q, K, V, O$, we only need to maintain states[3] $\alpha_R^{(t)}, \alpha_L^{(t)}, c_R^{(t)}, c_L^{(t)}$ from (9) and (10), resulting in $\Theta(Nd)$ additional memory. All other intermediate values are only materialized in on-chip SRAM, fusing all operations between the $\alpha$ (and similarly $c$) variables. For instance, from the above update equations, the computation of $\alpha_L^{(t)}$ from $\alpha_R^{(t)}$ is given by

$$\alpha_{L,jkv}^{(t)} = \text{softmax}_i\left(\frac{\sum_v \alpha_{R,kjv}^{(t)} \overline{K}_{kiv}}{c_{R,kj}^{(t)}}\right) \overline{K}_{kiv},$$

---

[3]The $\alpha$ and $c$ variables can share the same memory location as those corresponding to $R$ can be derived from $L$ (and vice-versa).

```python
def al_cl_kernel(aR, cR, Kb): # Computes aL, cL from aR, cR
    R = softmax(bmm(aR, Kb.transpose(1, 2)) / cR[:, :, None], dim=2)
    cL = sum(R * log(R), dim=2).transpose(0, 1)
    aL = bmm(R, Kb).transpose(0, 1)
    return aL, cL

def ar_cr_kernel(aL, cL, Qb): # Computes aR, cR from aL, cL
    L = softmax(bmm(aL, Qb.transpose(1, 2)) - cL[:, :, None], dim=1)
    cR = sum(L, dim=2).transpose(0, 1)
    aR = bmm(L, Qb).transpose(0, 1)
    return aR, cR

def al_y_cl_kernel(aR, cR, Kb, Vb): # Fuse al_cl_kernel + Monarch matmul 1st step
    R = softmax(bmm(aR, Kb.transpose(1, 2)) / cR[:, :, None], dim=2)
    cL = sum(R * log(R), dim=2).transpose(0, 1)
    aL = bmm(R, Kb).transpose(0, 1)
    y = bmm(R, Vb).transpose(0, 1)
    return aL, y, cL

def z_kernel(aL, y, cL, Qb): # Monarch matmul 2nd step
    L = softmax(bmm(Qb, aL.transpose(1, 2)) - cL[:, None, :], dim=2)
    z = bmm(L, y).transpose(0, 1)
    return z

def monarch_attention(Q, K, V, T): # Q, K, V: (N, d), T: number of steps
    Qb = Q.reshape(m, b, d).transpose(0, 1)
    Kb = K.reshape(m, b, d)
    Vb = V.reshape(m, b, d)
    aR = Q.reshape(m, b, d)
    cR = ones(m, b)

    for t in range(T-1):
        aL, cL = al_cl_kernel(aR, cR, Kb)
        aR, cR = ar_cr_kernel(aL, cL, Qb)

    aL, y, cL = al_y_cl_kernel(aR, cR, Kb, Vb)
    z = z_kernel(aL, y, cL, Qb)
    o = z.reshape(N, d)
    return o
```

Figure 4: **Python-like code for** `MonarchAttention`**.** Each kernel materializes all intermediate arrays in SRAM to reduce data movement.

which can be seen as a batched attention computation, meaning we can implement a `FlashAttention`-like kernel to reduce IO between HBM and SRAM. However, several aspects of this computation make it particularly IO-efficient. Besides the fact that $\overline{\boldsymbol{K}}$ acts as both the $\boldsymbol{K}$ and $\boldsymbol{V}$ matrices in (3), the effective sequence length is $\sqrt{N}$. This eliminates the need for tiling along the sequence length, except for very long sequences having $\Theta(\sqrt{N}d) > S$, where $S$ is the size of on-chip SRAM. This means that we have an optimal IO complexity of $\Theta(Nd)$ for a single call, as opposed to the worst-case $O(N^2d^2/S)$ complexity of `FlashAttention`. The computation of $\boldsymbol{\alpha}_R^{(t)}$ from $\boldsymbol{\alpha}_L^{(t)}$, as well as the Monarch matmul, can be written in a similar fashion. Based on this, `MonarchAttention` not only achieves significant speed-up over `FlashAttention` for longer sequences, but also for shorter ones. Python-like code for `MonarchAttention` is given in Figure 4.

## 4 Experiments

In this section, we evaluate the zero-shot performance (no additional training) of `MonarchAttention` for converting pre-trained/fine-tuned Transformer attention layers to sub-quadratic attention in four different model/task settings. We compare with previous low-rank attention methods (Katharopoulos et al., 2020; Choromanski et al., 2021; Xiong et al., 2021; Qin et al., 2022); see Appendix D.1 for

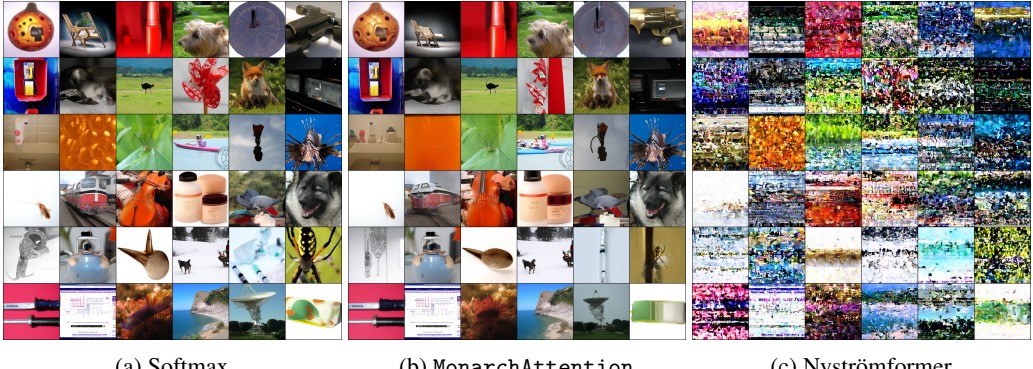

|  (a) Softmax | (b) `MonarchAttention` | (c) Nyströmformer |

Figure 5: **Visual quality of generated images for zero-shot conversion of attention layers.** Example images generated by with softmax (left), `MonarchAttention` (middle), and Nyströmformer (right). Only the first half of the attention layers of DiT are replaced.

more details on the baselines. We specifically exclude low-rank methods with learnable components (Wang et al., 2020; Zhang et al., 2024), since we are focused on the zero-shot setting, as well as sparsity/LSH-based approaches (Kitaev et al., 2020; Daras et al., 2020; Chen et al., 2021; Han et al., 2024), since these do not admit efficient implementations on current GPUs. We also benchmark our fast implementation of `MonarchAttention`, comparing with `FlashAttention-2` (Dao, 2024).

**Image Classification with Vision Transformer.** We convert all 12 attention layers, each having 12 heads with sequence length $N = 197$ and head dimension $d = 64$, of the 87M parameter ViT-B (Dosovitskiy et al., 2021) that has been pre-trained on ImageNet-21K (Deng et al., 2009) and fine-tuned on ImageNet-1K (Russakovsky et al., 2015) for image classification. To evaluate the performance at different FLOP counts, we vary the number of steps $T \in \{1, 2, 3\}$ for `MonarchAttention`, and vary the rank for Performer and Nyströmformer; see Appendix D.2 for more details on the set-up. The results are shown in the left panel of Figure 2. `MonarchAttention` achieves significant improvement over other baselines – compared to the original softmax attention, `MonarchAttention` loses only 5% accuracy to reduce attention FLOPs by 80%, or matches the performance to reduce attention FLOPs by 50%.

**Question Answering with Encoder-Only Transformer.** We convert the initial 4 and final 4 layers of the 12 attention layers, each having 12 heads with sequence length $N = 384$ and head dimension $d = 64$, of the 125M parameter RoBERTa-B (Liu et al., 2019) that has been pre-trained on a large English corpus and fine-tuned on SQuAD1.1 (Rajpurkar et al., 2016) for question answering. To evaluate the performance at different FLOP counts, we fix $T = 1$ and vary the block size $b$ for `MonarchAttention`, and vary the rank for Performer and Nyströmformer; see Appendix D.3 for more details on the set-up. The results are shown in the right panel of Figure 2. Once again, `MonarchAttention` achieves significant improvement over other baselines – compared to the original softmax attention, `MonarchAttention` loses only 10 points in F1 score to reduce attention FLOPs by 60%, or matches the performance to reduce attention FLOPs by 35%. We additionally provide a per-layer ablation study in Appendix E.2 to understand the impact of converting different layers in this setting.

**Summarization with Encoder-Decoder Transformer.** We convert all 6 attention layers, each having 12 heads with head dimension $d = 64$, in the encoder of the 139M parameter BART-B (Lewis et al., 2020) that has been pre-trained on a large English corpus and fine-tuned on BookSum-chapters (Kryściński et al., 2022) for summarization. We only convert the encoder model and leave the decoder intact. To evaluate the benefits of sub-quadratic attention for processing longer sequences, we truncate the text to be summarized to various sequence lengths $N$ for each method, with $T \in \{2, 3\}$ for `MonarchAttention` depending on the sequence length; see Appendix D.4 for more details on the set-up. The results are shown in Figure 3. We see that `MonarchAttention` achieves a strictly better ROUGE score (Lin, 2004) vs. FLOPs tradeoff than even softmax attention, due to accurate and efficient processing of longer sequences. In particular, the $N = 8192$ `MonarchAttention` model

Table 1: **Quantitative results for zero-shot conversion of attention layers for image generation.** We report FID and sFID (using the original softmax attention model as reference) of DiT when replacing all or half of the attention layers.

| Layers Replaced | Method | Total Attention FLOPs ($10^9$) | FID ($\downarrow$) | sFID ($\downarrow$) |
|---|---|---|---|---|
| – | Softmax | 8.46 | – | – |
| All | Nyströmformer | 3.30 | 5.97 | 13.47 |
| | MonarchAttention | 3.44 | **2.82** | **5.09** |
| First Half | Nyströmformer | 5.88 | 8.17 | 19.01 |
| | MonarchAttention | 5.95 | **0.39** | **0.66** |
| Second Half | Nyströmformer | 5.88 | 6.76 | 13.58 |
| | MonarchAttention | 5.95 | **1.98** | **3.36** |

improves on the $N = 2048$ softmax attention model by $0.75$ on ROUGE-1 and $0.5$ on ROUGE-L with slightly fewer FLOPs, while the $N = 8192$ Nyströmformer model with similar FLOPs does strictly worse than softmax. Although we are mainly focused on the zero-shot setting, we also investigate the post-swap fine-tuning performance and stability of MonarchAttention on this task in Appendix E.3.

**Image Generation with Diffusion Transformer.** We convert a subset of the 28 attention layers, each having 16 heads with sequence length $N = 256$ and head dimension $d = 72$, of the 675M parameter DiT-XL (Peebles and Xie, 2023) that has been trained on ImageNet (Deng et al., 2009). We consider replacing either all layers, the first $14$ layers, or the last $14$ layers, and fix $T = 3$ for MonarchAttention; see Appendix D.5 for more details on the set-up. Examples of generated images with each method for replacing the first 14 layers are shown in Figure 5, where MonarchAttention produces clear images resembling those of softmax attention, while the Nyströmformer ones are extremely noisy. We also quantitatively evaluate the (s)FID scores (Heusel et al., 2017) of MonarchAttention compared with Nyströmformer, using images generated with the original softmax attention model as reference – the results are reported in Table 1. MonarchAttention noticeably outperforms Nyströmformer with similar FLOP count. In particular, using MonarchAttention in the first half of the DiT layers results in extremely small FID and sFID from the softmax attention model's images, while reducing FLOPs by nearly $30\%$. We provide a more fine-grained version of Table 1 for MonarchAttention in Appendix E.4 to investigate the impact of replacing different quarters of the DiT model.

**Node Classification with Graph Transformer.** Beyond image and language tasks, we also evaluate the performance of MonarchAttention for converting attention in Graph Transformers on a graph node classification task in Appendix E.5.

**Benchmarking MonarchAttention.** Finally, we validate that the computational/IO complexity reduction achieved by MonarchAttention translates into actual speed-ups on the NVIDIA A40, a modern GPU. We implement the pseudo-code described in Section 3 as four separate Triton kernels and compare it against the fastest available implementations of FlashAttention-2 – either the Triton implementation or PyTorch's scaled_dot_product_attention, which calls the CUDA backend for FlashAttention-2. Using a fixed batch size $E$, number of heads $H$, and head dimension $d$, we sweep the input sequence length $N$, generate random $\boldsymbol{Q}, \boldsymbol{K}, \boldsymbol{V}$ matrices, and compare the run-times of FlashAttention-2 and MonarchAttention (with $b = \sqrt{N}$ and $T = 1$) in Figure 6 (left). As sequence length increases, MonarchAttention consistently outperforms FlashAttention-2, notably achieving up to $8.2\times$ speed-up with $N = 16384$. To highlight gains for shorter sequences, we implement MonarchAttention as a single fully-fused Triton kernel, with a single thread block computing a single head. For fixed sequence length $N = 256$, number of heads $H$, and head dimension $d$, we sweep the batch size $E$ and compare the run-time of the fully-fused MonarchAttention kernel against FlashAttention-2 in Figure 6 (right). With smaller batch sizes, we have low utilization of hardware, since we compute a single head with a single thread block. However, as we increase the batch size, MonarchAttention achieves up to $1.4\times$ speed-up over FlashAttention-2.

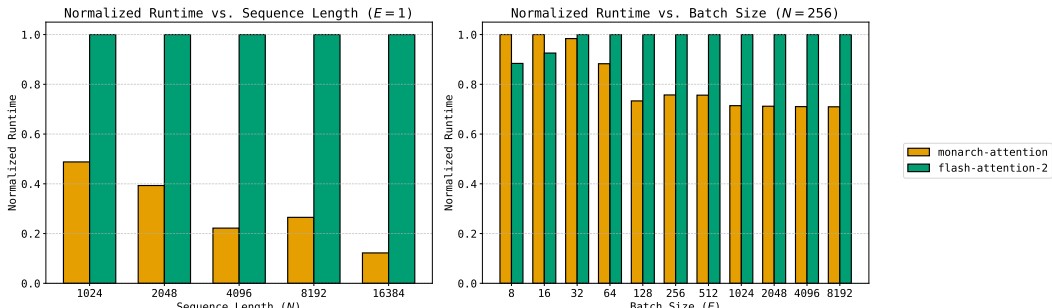

Figure 6: **Run-times of** `MonarchAttention` **and** `FlashAttention-2` **across various sequence lengths.** Normalized runtime ($1 = $ slowest, $0 = $ fastest) of `MonarchAttention` and `FlashAttention-2` on NVIDIA A40 GPU. *Left*: sweep of sequence length $N$ with $E = 1$, $H = 12$, and $d = 64$. *Right*: sweep of batch size $E$ with $N = 256$, $H = 12$, and $d = 64$.

## 5 Conclusion

To conclude, we discuss several limitations and future directions for this work.

**Autoregressive Transformers.** In this paper, we have primarily focused on zero-shot approximation of attention in encoder-based Transformers, yet most generative language models of interest today are decoder-based autoregressive Transformers. One limitation of `MonarchAttention` is that it fundamentally cannot be applied to autoregressive generation, since there is no "attention matrix" to approximate – instead, single queries are passed through the attention mechanism in a streaming fashion, and discarded after each decode step. Instead, `MonarchAttention` could be used during prefill to efficiently compute keys/values for a provided prefix (e.g., long prompt or context). Moreover, `MonarchAttention` can be applied to emerging non-autoregressive paradigms such as diffusion language models (DLMs) to accelerate language generative models.

**Accelerating Training.** While we have presented `MonarchAttention` as a direct replacement for softmax attention with no additional training, it can also accelerate pre-training or fine-tuning as well. Although `MonarchAttention` cannot be applied at *inference* time for autoregressive Transformers, it can be applied to *training* such models. However it is not obvious how to efficiently incorporate the causal mask into `MonarchAttention`.

**Beyond Monarch & Softmax.** Finally, although `MonarchAttention` is specifically designed for approximating softmax attention via Monarch, this idea can be generalized to other (possibly dynamic) structured matrix classes, as well as other nonlinear mappings. Monarch, while efficient, allocates the same number of parameters to each uniformly sized block. Yet, in practice, different regions of the attention matrix may require more fine-grained approximation. On the other hand, $\alpha$-entmax mappings (Peters et al., 2019), which generalize softmax, can produce sparser attention matrices, thereby resulting in better approximations via block rank-one type matrices.

## Acknowledgement

LB and CY were supported in part by NSF CAREER award CCF-1845076, NSF award CCF-2331590, and an Intel Early Career award. LB was also supported by the University of Michigan Crosby award. CY and AX were supported by NSF CCF 312842. PA and CL were supported in part by COGNISENSE, one of seven centers in JUMP 2.0, a Semiconductor Research Corporation (SRC) program sponsored by DARPA. We thank Samet Oymak (University of Michigan) for discussion and use of computational resources provided by an Amazon Research Award on Foundation Model Development.

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

# Appendix

## A   Equivalence of Softmax Definitions

Consider the optimization problem in (2):

$$\max_{\boldsymbol{a}} \quad f(\boldsymbol{a}) := \sum_i \boldsymbol{a}_i \boldsymbol{z}_i - \sum_i \boldsymbol{a}_i \log \boldsymbol{a}_i \quad \text{s.t.} \quad \boldsymbol{a}_i \geq 0, \quad \sum_i \boldsymbol{a}_i = 1.$$

From the KKT stationarity condition, we have

$$\frac{\partial}{\partial \boldsymbol{a}_i} \left( f(\boldsymbol{a}) + \lambda \left( 1 - \sum_i \boldsymbol{a}_i \right) + \sum_i \boldsymbol{\mu}_i \boldsymbol{a}_i \right) = 0 \implies \boldsymbol{z}_i - (1 + \log \boldsymbol{a}_i) - \lambda + \boldsymbol{\mu}_i = 0,$$

where $\lambda \in \mathbb{R}, \boldsymbol{\mu} \in \mathbb{R}^N$ are dual variables. From complementary slackness $\boldsymbol{a}_i \boldsymbol{\mu}_i = 0$ and the fact that $\log \boldsymbol{a}_i$ is not defined for $\boldsymbol{a}_i = 0$, we must have $\mu_i = 0$, which gives

$$\log \boldsymbol{a}_i = \boldsymbol{z}_i - \lambda - 1 \implies \boldsymbol{a}_i = \exp(\boldsymbol{z}_i)/\exp(\lambda + 1).$$

Finally, from the constraint $\sum_i \boldsymbol{a}_i = 1$, we must have $\exp(\lambda + 1) = \sum_j \exp(\boldsymbol{z}_j)$, which gives the form of softmax in (1).

## B   Monarch Background

We provide an example of the transpose permutation $\boldsymbol{P}$ in Section 2. Recall that applying $\boldsymbol{P} \in \mathbb{R}^{N \times N}$ to a vector $\boldsymbol{x} \in \mathbb{R}^N$ corresponds to row-major reshaping $\boldsymbol{x}$ to $\mathbb{R}^{m \times b}$, transposing to $\mathbb{R}^{b \times m}$, then row-major flattening back to $\mathbb{R}^N$. This is equivalent to applying a permutation matrix whose $(i+1)$-th row is given by $\boldsymbol{e}_{\sigma(i)+1}$ where

$$\sigma(i) = b \cdot (i \bmod m) + \left\lfloor \frac{i}{m} \right\rfloor, \quad i \in \{0, \ldots, N-1\}.$$

As an illustrative example, let $N = 6$, $b = 3$, and $m = 2$. The action of $\boldsymbol{P}$ is given by the following steps:

$$\begin{bmatrix} 1 \\ 2 \\ 3 \\ 4 \\ 5 \\ 6 \end{bmatrix} \xrightarrow{\text{reshape } 2 \times 3} \begin{bmatrix} 1 & 2 & 3 \\ 4 & 5 & 6 \end{bmatrix} \xrightarrow{\text{transpose}} \begin{bmatrix} 1 & 4 \\ 2 & 5 \\ 3 & 6 \end{bmatrix} \xrightarrow{\text{flatten}} \begin{bmatrix} 1 \\ 4 \\ 2 \\ 5 \\ 3 \\ 6 \end{bmatrix}.$$

In matrix form, we have

$$\boldsymbol{P} = \begin{bmatrix} 1 & 0 & 0 & 0 & 0 & 0 \\ 0 & 0 & 0 & 1 & 0 & 0 \\ 0 & 1 & 0 & 0 & 0 & 0 \\ 0 & 0 & 0 & 0 & 1 & 0 \\ 0 & 0 & 1 & 0 & 0 & 0 \\ 0 & 0 & 0 & 0 & 0 & 1 \end{bmatrix}.$$

## C   Details for `MonarchAttention`

### C.1   Updates

**Derivatives.**   We evaluate $f$ with $\boldsymbol{A} = \boldsymbol{M}$ as a Monarch matrix. Using (5), we have

$$\begin{aligned} f(\boldsymbol{M}; \boldsymbol{Q}, \boldsymbol{K}) &= \sum \boldsymbol{L}_{jkl} \boldsymbol{R}_{kji} \overline{\boldsymbol{Q}}_{jlv} \overline{\boldsymbol{K}}_{kiv} - \sum \boldsymbol{L}_{jkl} \boldsymbol{R}_{kji} \log(\boldsymbol{L}_{jkl} \boldsymbol{R}_{kji}) \\ &= \sum \boldsymbol{L}_{jkl} \boldsymbol{R}_{kji} \overline{\boldsymbol{Q}}_{jlv} \overline{\boldsymbol{K}}_{kiv} - \sum \boldsymbol{L}_{jkl} \boldsymbol{R}_{kji} \log \boldsymbol{R}_{kji} - \sum \boldsymbol{R}_{kji} \boldsymbol{L}_{jkl} \log \boldsymbol{L}_{jkl}. \end{aligned}$$

The derivatives of $f$ w.r.t. each factor are given by

$$\frac{\partial f(\boldsymbol{M};\boldsymbol{Q},\boldsymbol{K})}{\partial \boldsymbol{L}_{jkl}} = \boldsymbol{\beta}_{L,jkl} - \boldsymbol{c}_{L,jk} - (1 + \log \boldsymbol{L}_{jkl})\boldsymbol{\gamma}_{L,jk}, \tag{11}$$

$$\frac{\partial f(\boldsymbol{M};\boldsymbol{Q},\boldsymbol{K})}{\partial \boldsymbol{R}_{kji}} = \boldsymbol{\beta}_{R,kji} - \boldsymbol{\gamma}_{R,kj} - (1 + \log \boldsymbol{R}_{kji})\boldsymbol{c}_{R,kj}, \tag{12}$$

where

$$\boldsymbol{\beta}_{L,jkl} = \sum_v \overline{\boldsymbol{Q}}_{jlv} \sum_i (\boldsymbol{R}_{kji}\overline{\boldsymbol{K}}_{kiv}), \quad \boldsymbol{c}_{L,jk} = \sum_i \boldsymbol{R}_{kji} \log \boldsymbol{R}_{kji}, \quad \boldsymbol{\gamma}_{L,jk} = \sum_i \boldsymbol{R}_{kji},$$

$$\boldsymbol{\beta}_{R,kji} = \sum_v \overline{\boldsymbol{K}}_{kiv} \sum_l (\boldsymbol{L}_{jkl}\overline{\boldsymbol{Q}}_{jlv}), \quad \boldsymbol{\gamma}_{R,kj} = \sum_l \boldsymbol{L}_{jkl} \log \boldsymbol{L}_{jkl}, \quad \boldsymbol{c}_{R,kj} = \sum_l \boldsymbol{L}_{jkl}.$$

We derive updates for each factor based on maximizing $f$ with the other factor fixed.

***L* update.** First, we fix $\boldsymbol{R} \in \Delta^{m \times b \times b}$ and consider

$$\max_{\boldsymbol{L}} \quad f(\boldsymbol{M};\boldsymbol{Q},\boldsymbol{K}) \quad \text{s.t.} \quad \boldsymbol{L}_{jkl} \geq 0, \quad \sum_k \boldsymbol{L}_{jkl} = 1.$$

From the KKT stationarity condition, we have

$$\frac{\partial}{\partial \boldsymbol{L}_{jkl}} \left( f(\boldsymbol{M};\boldsymbol{Q},\boldsymbol{K}) + \sum \boldsymbol{\lambda}_{L,jl} \left( 1 - \sum_k \boldsymbol{L}_{jkl} \right) + \sum \boldsymbol{\mu}_{L,jkl}\boldsymbol{L}_{jkl} \right) = 0$$

where $\boldsymbol{\lambda}_L \in \mathbb{R}^{b \times m}, \boldsymbol{\mu}_L \in \mathbb{R}^{b \times m \times m}$ are dual variables. Along with (11), we have

$$\boldsymbol{\beta}_{L,jkl} - \boldsymbol{c}_{L,jk} - (1 + \log \boldsymbol{L}_{jkl})\boldsymbol{\gamma}_{L,jk} - \boldsymbol{\lambda}_{L,jl} + \boldsymbol{\mu}_{L,jkl} = 0.$$

Now, from complementary slackness $\boldsymbol{\mu}_{L,jkl}\boldsymbol{L}_{jkl} = 0$ and the fact that $\log \boldsymbol{L}_{jkl}$ is not defined for $\boldsymbol{L}_{jkl} = 0$, we must have $\boldsymbol{\mu}_{L,jkl} = 0$. Moreover, since $\boldsymbol{R} \in \Delta^{m \times b \times b}$, we have $\boldsymbol{\gamma}_{L,jk} = 1$. Altogether, we have

$$\log \boldsymbol{L}_{jkl} = \boldsymbol{\beta}_{L,jkl} - \boldsymbol{c}_{L,jk} - \boldsymbol{\lambda}_{L,jl} - 1 \implies \boldsymbol{L}_{jkl} = \frac{\exp(\boldsymbol{\beta}_{L,jkl} - \boldsymbol{c}_{L,jk})}{\exp(\boldsymbol{\lambda}_{L,jl} + 1)}.$$

Finally, from the constraint $\sum_k \boldsymbol{L}_{jkl} = 1$, we must have $\exp(\boldsymbol{\lambda}_{L,jl} + 1) = \sum_k \exp(\boldsymbol{\beta}_{L,jkl} - \boldsymbol{c}_{L,jk})$, which gives the final closed form update:

$$\boldsymbol{L} = \text{softmax}_k(\boldsymbol{Z}_L), \quad \boldsymbol{Z}_{L,jkl} = \boldsymbol{\beta}_{L,jkl} - \boldsymbol{c}_{L,jk},$$

where $\text{softmax}_k$ is applied along the $k$ index dimension.

***R* update.** Similarly, we fix $\boldsymbol{L} \in \Delta^{b \times m \times m}$ and consider

$$\max_{\boldsymbol{R}} \quad f(\boldsymbol{M};\boldsymbol{Q},\boldsymbol{K}) \quad \text{s.t.} \quad \boldsymbol{R}_{kji} \geq 0, \quad \sum_i \boldsymbol{R}_{kji} = 1.$$

From the KKT stationarity condition, we have

$$\frac{\partial}{\partial \boldsymbol{R}_{kji}} \left( f(\boldsymbol{M};\boldsymbol{Q},\boldsymbol{K}) + \sum \boldsymbol{\lambda}_{R,kj} \left( 1 - \sum_i \boldsymbol{R}_{kji} \right) + \sum \boldsymbol{\mu}_{R,kji}\boldsymbol{R}_{kji} \right) = 0$$

where $\boldsymbol{\lambda}_R \in \mathbb{R}^{m \times b}, \boldsymbol{\mu}_R \in \mathbb{R}^{m \times b \times b}$ are dual variables. Along with (12), we have

$$\boldsymbol{\beta}_{R,kji} - \boldsymbol{\gamma}_{R,kj} - (1 + \log \boldsymbol{R}_{kji})\boldsymbol{c}_{R,kj} - \boldsymbol{\lambda}_{R,kj} + \boldsymbol{\mu}_{R,kji} = 0.$$

As before, from complementary slackness $\boldsymbol{\mu}_{R,kji}\boldsymbol{R}_{kji} = 0$ and the fact that $\log \boldsymbol{R}_{kji}$ is not defined for $\boldsymbol{R}_{kji} = 0$, we must have $\boldsymbol{\mu}_{R,kji} = 0$. Altogether, we have

$$\log \boldsymbol{R}_{kji} = \frac{\boldsymbol{\beta}_{R,kji} - \boldsymbol{\gamma}_{R,kj} - \boldsymbol{\lambda}_{R,kj}}{\boldsymbol{c}_{R,kj}} - 1 \implies \boldsymbol{R}_{kji} = \frac{\exp(\boldsymbol{\beta}_{R,kji}/\boldsymbol{c}_{R,kj})}{\exp(\boldsymbol{\gamma}_{R,kj}/\boldsymbol{c}_{R,kj} + \boldsymbol{\lambda}_{R,kj}/\boldsymbol{c}_{R,kj} + 1)}.$$

Finally, from the constraint $\sum_i \boldsymbol{R}_{kji} = 1$, we must have $\exp(\boldsymbol{\gamma}_{R,kj}/\boldsymbol{c}_{R,kj} + \boldsymbol{\lambda}_{R,kj}/\boldsymbol{c}_{R,kj} + 1) = \sum_i \exp(\boldsymbol{\beta}_{R,kji}/\boldsymbol{c}_{R,kj})$, which gives the final closed form update:

$$\boldsymbol{R} = \text{softmax}_i(\boldsymbol{Z}_R), \quad \boldsymbol{Z}_{R,kji} = \boldsymbol{\beta}_{R,kji}/\boldsymbol{c}_{R,kj},$$

where $\text{softmax}_i$ is applied along the $i$ index dimension.

## C.2   Naïve Algorithm

We provide pseudo-code for the naive version of `MonarchAttention` in Figure 7. This is a direct implementation of alternating maximization for finding the Monarch factors, where we highlight the choices to (1) initialize $L$ to be block identity, and (2) update $R$ before $L$ in each iteration. We include this here for completeness, so that readers can directly compare this code with the algorithmic steps. The implementation in Figure 4 is a reorganization of Figure 7 where $L, R$ are not materialized in HBM, resulting in `FlashAttention`-like kernels.

```
1   # Q: array of size (N, d)
2   # K: array of size (N, d)
3   # V: array of size (N, d)
4   # T: number of steps
5
6   def monarch_attention(Q, K, V, T):
7       L = stack(b * [eye(m)])
8       Qb = einshape("(lj)v->jlv", Q, j=b)
9       Kb = einshape("(ki)v->kiv", K, i=b)
10
11      # Alternating maximization for L, R
12      for t in range(T):
13          # R update
14          aR = einsum("jkl,jlv->kjv", L, Qb)
15          bR = einsum("kjv,kiv->kji", aR, Kb)
16          cR = einsum("jkl->kj", L)
17          R = softmax(bR / cR[:, :, None], axis=2)
18
19          # L update
20          aL = einsum("kji,kiv->jkv", R, Kb)
21          bL = einsum("jkv,jlv->jkl", aL, Qb)
22          cL = einsum("kji->jk", R * log(R))
23          L = softmax(bL - cL[:, :, None], axis=1)
24
25      # Monarch multiply
26      Vb = einshape("(ki)v->kiv", V, i=b)
27      Y = einsum("kji,kiv->jkv", R, Vb)
28      Z = einsum("jkl,jkv->ljv", L, Y)
29      O = einshape("ljv->(lj)v", Z)
30
31      return O
```

Figure 7: Naïve implemention of `MonarchAttention`

## C.3   Padding and Masking

In practice, the sequence length $N$ may not be divisible by the desired block size $b$. In such cases, we round the number of blocks $m$ to $m' = \lceil N/b \rceil$, and set the new sequence length $N' = m'b$, post-padding $Q, K, V$ to have $N'$ rows. However, we need to take special care that the final $N' - N$ columns of the padded Monarch attention matrix $M \in \Delta^{N' \times N'}$ are zero, since these correspond to padded rows of $V$. This is also an issue when batched sequences of different lengths are padded to a maximum length to avoid dynamic resizing.

From (5), it is clear that to set all columns of $M$ beyond the $N$-th column to zero, it is sufficient to set $R_{kji}$ to zero whenever $b(k-1) + i > N$. Thus, we simply form the mask $\omega \in \mathbb{R}^{m' \times b \times b}$ given by

$$\omega_{kji} = \begin{cases} 0 & b(k-1) + i \leq N \\ -\infty & \text{otherwise,} \end{cases}$$

which we then add to $Z_R$ before softmax in (7). We can also pre-pad the sequence, which would be change the above condition to $b(k-1) + i \geq N' - N$.

# D   Experimental Details

## D.1   Baselines

We describe the baselines used in Section 4.

- `linear-attention` (Katharopoulos et al., 2020) approximates $\exp(\boldsymbol{q}^\top \boldsymbol{k}) \approx \phi(\boldsymbol{q})^\top \phi(\boldsymbol{k})$ where $\phi : \mathbb{R}^d \to \mathbb{R}^r$ is a kernel feature map, resulting in a rank $r$ approximation to softmax attention:
$$\mathrm{softmax}(\boldsymbol{Q}\boldsymbol{K}^\top) \approx \frac{\phi(\boldsymbol{Q})\phi(\boldsymbol{K})^\top}{\phi(\boldsymbol{Q})\phi(\boldsymbol{K})^\top \mathbf{1}_N \mathbf{1}_N^\top}.$$
Katharopoulos et al. (2020) propose the map $\phi(\boldsymbol{x}) = 1 + \mathrm{elu}(\boldsymbol{x})$ with $r = d$ where elu is the exponential linear unit applied element-wise.

- `performer` (Choromanski et al., 2021) is a linear attention method using the fact that
$$\exp(\boldsymbol{q}^\top \boldsymbol{k}) = \mathbb{E}_{\boldsymbol{\omega} \sim \mathcal{N}(\mathbf{0}, \boldsymbol{I}_d)}\left[ \exp\left( \boldsymbol{\omega}^\top \boldsymbol{q} - \frac{\|\boldsymbol{q}\|^2}{2} \right) \exp\left( \boldsymbol{\omega}^\top \boldsymbol{k} - \frac{\|\boldsymbol{k}\|^2}{2} \right) \right]$$
to construct a random kernel feature map
$$\phi(\boldsymbol{x}) = \frac{1}{\sqrt{r}} \exp\left( -\frac{\|\boldsymbol{x}\|^2}{2} \right) \left[ \exp(\boldsymbol{\omega}_1^\top \boldsymbol{x}) \quad \dots \quad \exp(\boldsymbol{\omega}_r^\top \boldsymbol{x}) \right]^\top,$$
where $\boldsymbol{\omega}_1, \dots, \boldsymbol{\omega}_r \overset{iid}{\sim} \mathcal{N}(\mathbf{0}, \boldsymbol{I}_d)$.

- `cosformer` (Qin et al., 2022) is a linear attention method utilizing position-dependent kernel feature maps of the form
$$\phi_i(\boldsymbol{x}) = \left[ \sin\left( \frac{\pi i}{2N} \right) \mathrm{relu}(\boldsymbol{x}_i) \quad \cos\left( \frac{\pi i}{2N} \right) \mathrm{relu}(\boldsymbol{x}_i) \right], \ \forall i \in [N],$$
which produces a rank $r = 2d$ approximation.

- `nystromformer` (Xiong et al., 2021) computes landmark $\tilde{\boldsymbol{Q}}, \tilde{\boldsymbol{K}} \in \mathbb{R}^{r \times d}$ from $\boldsymbol{Q}, \boldsymbol{K}$ by averaging $N/r$ consecutive spans of rows, which are used to approximate softmax attention via the quadrature method:
$$\tilde{\boldsymbol{F}} = \mathrm{softmax}(\boldsymbol{Q}\tilde{\boldsymbol{K}}^\top), \ \tilde{\boldsymbol{B}} = \mathrm{softmax}(\tilde{\boldsymbol{Q}}\boldsymbol{K}^\top), \ \tilde{\boldsymbol{A}} = \mathrm{softmax}(\tilde{\boldsymbol{Q}}\tilde{\boldsymbol{K}}^\top)$$
$$\mathrm{softmax}(\boldsymbol{Q}\boldsymbol{K}^\top) \approx \tilde{\boldsymbol{F}}\tilde{\boldsymbol{A}}^+ \tilde{\boldsymbol{B}},$$
where $\tilde{\boldsymbol{A}}^+$ denotes the pseudoinverse of $\tilde{\boldsymbol{A}}$, producing a rank $r$ approximation.

## D.2   Image Classification with ViT

The ViT-B model fine-tuned on ImageNet-21K is retrieved from the Hugging Face `transformers` library (Wolf et al., 2019) as `google/vit-base-patch16-224`. The ImageNet-1K evaluation dataset is retrieved from the Hugging Face `datasets` library (Lhoest et al., 2021) as `imagenet-1k` using the `validation` split. We vary the following hyperparameters:

- `monarch-attention`: $b = 14$ and $T \in \{1, 2, 3\}$
- `performer`: $r \in \{16, 32, 48, 64, 80, 96\}$
- `nystromformer`: $r \in \{16, 24, 32, 40\}$

The choice of $b = 14$ for `monarch-attention` is due to the fact that images are patched in a $14 \times 14$ grid. We also apply post-padding as described in Appendix C.3 for the `[CLS]` token, since it is appended at the end of the sequence.

## D.3   Question Answering with RoBERTa

The RoBERTa-B model fine-tuned on SQuAD1.1 is retrieved from the Hugging Face `transformers` library as `csarron/roberta-base-squad-v1`. The SQuAD1.1 evaluation dataset is retrieved from the Hugging Face `datasets` library as `squad` using the `validation` split. For evaluation, we truncate and pad to sequence length of 384. We vary the following hyperparameters:

Table 2: Hyperparameters used for BART summarization.

| Sequence Length | Method | $(b, T)$ | $r$ | Total Attention FLOPs ($10^9$) |
|---|---|---|---|---|
| 1024 | Softmax | – | – | 9.66 |
| | Nyströmformer | – | 64 | 1.93 |
| | MonarchAttention | (32, 3) | – | 1.96 |
| 2048 | Softmax | – | – | 38.7 |
| | Nyströmformer | – | 80 | 4.41 |
| | MonarchAttention | (32, 2) | – | 3.93 |
| 4096 | Softmax | – | – | 155. |
| | Nyströmformer | – | 112 | 10.6 |
| | MonarchAttention | (64, 2) | – | 10.9 |
| 8192 | Softmax | – | – | 619. |
| | Nyströmformer | – | 160 | 35.0 |
| | MonarchAttention | (64, 2) | – | 31.4 |

- `monarch-attention`: $T = 1$ and $b \in \{24, 48, 96, 128\}$
- `performer`: $r \in \{32, 64, 96, 128, 160, 192\}$
- `nystromformer`: $r \in \{16, 32, 48, 64\}$

### D.4  Summarization with BART

The pre-trained BART-B model is retrieved from the Hugging Face `transformers` library as `facebook/bart-base`. The BookSum-chapters training/evaluation dataset is retrieved from the Hugging Face `datasets` library as `kmfoda/booksum` using the `train` and `validation` splits respectively. BART employs learned positional embeddings up to 1024 sequence length, and since we are interested in long-sequence summarization up to 8192 tokens, we linearly interpolate the encoder positional embeddings up to 8192 tokens, before fine-tuning on BookSum-chapters – we leave the decoder positional embeddings intact. We fine-tune for 5 epochs with batch size of 32 and learning rate of $10^{-4}$ using the Adam optimizer (Kingma and Ba, 2014) without weight decay, with the input and summary sequences truncated and padded to 8192 and 512 tokens respectively. For evaluation, we truncate the input sequence to the corresponding sequence length in Figure 3. The hyperparameters for each method across sequence lengths are shown in Table 2.

### D.5  Image Generation with DiT

The pre-trained model DiT-XL is retrieved from the Hugging Face `transformers` library as `facebook/DiT-XL-2-256`. Following Peebles and Xie (2023), we generate images using 32 sampling steps, a $2 \times 2$ patch size, and a classifier-free guidance scale of 1.5. We use the following hyperparameters:

- `monarch-attention`: $b = 16$ and $T = 3$
- `nystromformer`: $r = 32$

To create the images in Figure 5, we used a random seed of 0 input the *same* 36 random Gaussian samples into all three models. To obtain the results in Table 1, we again used a random seed of 0, and generated $50K$ images from each type of model, again using the *same* $50K$ random samples across all models.

## E  Additional Experimental Results

### E.1  Convergence of `MonarchAttention`

We empirically show that `MonarchAttention` converges very quickly in the ViT image classification setting described in Appendix D.2. We measure of the value of the softmax variational objective

$f(\boldsymbol{M}^{(T)}; \boldsymbol{Q}, \boldsymbol{K})$ as defined in (6) in the 0-th layer and 5-th head for the first few $T$ – the results are shown in Table 3. We see that by $T = 2$, the variational objective has already reached a stationary point, with even $T = 1$ closing most of the gap. This demonstrates how `MonarchAttention` is able to reach an accurate solution with even just one step.

Table 3: Convergence of $f(\boldsymbol{M}^{(T)}; \boldsymbol{Q}, \boldsymbol{K})$.

| $T$ | 0 | 1 | 2 | 3 |
|---|---|---|---|---|
| $f(\boldsymbol{M}^{(T)})$ | 418 | 937 | 940 | 940 |

## E.2 Layer Ablation for RoBERTa Question Answering

We investigate the impact of replacing different layers with `MonarchAttention` in the RoBERTa question answering setting described in Appendix D.3. The results are shown in Table 4.

Table 4: F1 score when replacing a given layer of RoBERTa with `MonarchAttention`.

| Layer | 0 | 1 | 2 | 3 | 4 | 5 | 6 | 7 | 8 | 9 | 10 | 11 |
|---|---|---|---|---|---|---|---|---|---|---|---|---|
| F1 ($\uparrow$) | 92.7 | 92.5 | 92.6 | 92.0 | 91.4 | 91.9 | 86.3 | 91.1 | 92.7 | 93.2 | 93.0 | 92.3 |

The initial and final layers are more amenable to replacement compared to the middle layers.

## E.3 Post-Swap Fine-Tuning for BART Summarization

We demonstrate the improved performance of `MonarchAttention` when fine-tuning on BART summarization post-replacement. We follow the setting and hyperparameters of Appendix D.4, but fine-tune with `MonarchAttention` or Nyströmformer instead of the standard softmax attention for sequence length $N = 8192$. The results are shown in Table 5, where we also copy the zero-shot results from the main body for reference.

Table 5: Post-swap fine-tuning vs zero-shot performance on BART summarization.

| Type | Method | Total Attention FLOPs ($10^9$) | ROUGE-1 ($\uparrow$) |
|---|---|---|---|
| – | Softmax | 618 | 35.57 |
| Fine-tuned | `MonarchAttention` | 31 | **35.38** |
|  | Nyströmformer | 35 | 33.94 |
| Zero-shot | `MonarchAttention` | 31 | **34.60** |
|  | Nyströmformer | 35 | 32.86 |

We see that `MonarchAttention` is stable under fine-tuning, and is indeed a strong approximation even beyond the zero-shot setting. In particular, we can almost fully close the gap with `MonarchAttention` to the original softmax model, while benefiting from more efficient fine-tuning with roughly 5% of the attention FLOPs.

## E.4 Layer Ablation for DiT Image Generation

We investigate the impact of replacing different quarters with `MonarchAttention` in the DiT image generation setting described in Appendix D.5. The results are shown in Table 6. We see that replacing earlier layers is more favorable than later layers, with the most degradation resulting from replacing the final layers of the DiT.

## E.5 Graph Node Classification with GraphGPS

We evaluate the efficacy of `MonarchAttention` beyond language and image modalities by considering zero-shot conversion of attention in the GPS graph Transformer (GPS-GT) (Rampášek et al.,

Table 6: (s)FID scores when replacing different quarters of DiT with `MonarchAttention`.

| Quarter | First | Second | Third | Fourth |
|---------|-------|--------|-------|--------|
| FID ($\downarrow$) | 0.07 | 0.21 | 0.34 | 1.35 |
| sFID ($\downarrow$) | 0.14 | 0.39 | 0.52 | 2.23 |

2022) on the Actor graph dataset (Pei et al., 2020) for node classification. In particular, we convert the attention layer in a single-layer 170K parameter GPS-GT with 2 heads, sequence length $N = 7600$, and head dimension $d = 48$. We train for 1500 steps with learning rate of $5 \times 10^{-4}$ using the Adam optimizer, and also evaluate on the train split[4]. To evaluate the performance at different FLOP counts, we vary the number of steps $T \in \{1, 2, 3, 4\}$ while fixing $b = 128$ for `MonarchAttention`, and vary the rank $r \in \{128, 192, 256, 320\}$ for Nyströmformer. The results are shown in Figure 8.

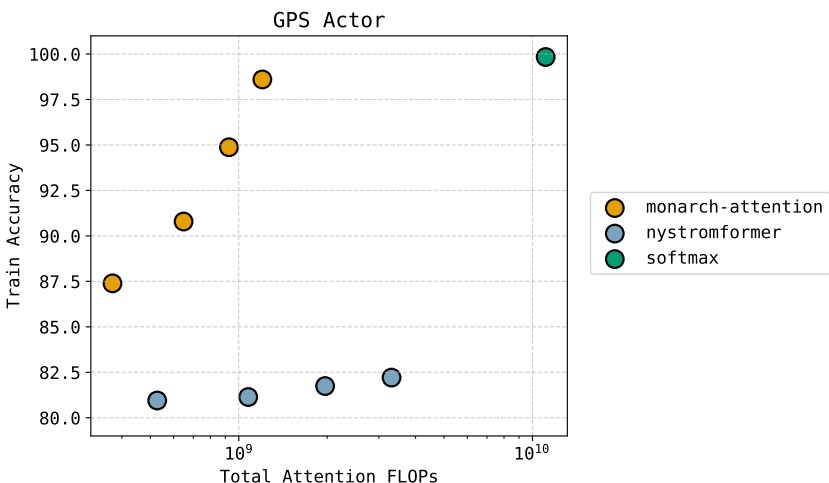

Figure 8: **Zero-shot conversion of attention layers for graph node classification.** We vary hyperparameters to evaluate training accuracy vs. total attention FLOPs across all layers for GPS on the Actor dataset.

`MonarchAttention` achieves a favorable quality-efficiency tradeoff, almost matching the softmax model quality with an order of magnitude less FLOPs. On the other hand, Nyströmformer achieves roughly 18% less accuracy with 2-3$\times$ more FLOPs than `MonarchAttention`.

---

[4]We evaluate on the train split due to low performance on the validation/test sets due to limited data/training from scratch. Since we are primarily interested in the post-train zero-shot approximation capabilities of `MonarchAttention`, this does not introduce bias in the evaluation.

