# OpenReview forum: "MonarchAttention: Zero-Shot Conversion to Fast, Hardware-Aware Structured Attention"
_NeurIPS.cc/2025/Conference — NeurIPS 2025 spotlight_

### Official Review · Reviewer_nTcP · 2025-06-27

**Clarity:** 4
**Significance:** 4
**Originality:** 4
**Rating:** 5
**Confidence:** 4

**Summary:**

The authors propose to approximate the softmax attention matrix with a Monarch matrix in order to decrease the computational cost from $\Theta(N^2 d)$ to $\Theta(N \sqrt{N} d)$, where $N$ is the sequence length and $d$ the embedding dimensionality. To that end, they cast attention as an optimization problem over candidate "attention" matrices, and derive an alternating maximization scheme that monotonously improves a Monarch approximation to the attention matrix in $\Theta(N \sqrt{N} d)$ without ever materializing the attention matrix. Empirically, the authors demonstrate runtime improvements over FlashAttention and much better attention approximation than linear transformers in image generation and language encoding. They demonstrate further that MonarchAttention achieves strictly better performance / FLOPs in summarization than truncated softmax.

**Questions:**

For well-definedness of the entropy terms, L and R must be constrained to
be positive. I don’t see this mentioned anywhere in Section 3 before subsection “Alternating Maximization with Constraints”.

The authors state: “When L is fixed, the objective is concave in R”. This seems to be a typo, since the negative entropy is convex.

The chosen $T$ is not always indicated in the experimental section, which initially gives the impression of it being quite large. I would suggest to explicitly state this for every setup to assure the reader no information is being hidden.

A discussion of how Monarch attention integrates with GQA and MLA could be a noteworthy addition.

**Ethical Concerns:**

["NO or VERY MINOR ethics concerns only"]

**Final Justification:**

The authors reaffirmed the strength of the paper and in particular elaborated on the empirical convergence rate of their optimization scheme. The paper is a strong contribution to the scientific community, I thus still recommend acceptance.

**Limitations:**

yes

**Quality:**

3

**Strengths And Weaknesses:**

Strengths:
- The paper is very clearly written.
- The related work is well-explored and contextualizes the author’s contribution.
- The experimental evaluation is extensive and convincing.
- The paper may have wide applicability, in particular for graph transformers and large reasoning models.

Weaknesses:
- The applications most impacted by a $\sqrt{N}$ acceleration of attention (e.g., graph attention and very long-context settings) are only explored to a limited extent.
- MonarchAttention only achieves improvements over vanilla attention if the number of optimization steps $T$ is in $o(\sqrt{N})$. Although in the experimental evaluations $T$ only reaches the modest size of $3$, the paper lacks insights on the convergence rate of the optimization problem as a function of $T$.

---

> ### Author Rebuttal · Authors · 2025-07-29
>
> We thank the reviewer for the positive reception and insightful points. We address each of the reviewer’s comments and questions in the following.
>
> *The applications most impacted by a sqrt(N) acceleration of attention (e.g., graph attention and very long-context settings) are only explored to a limited extent.*
>
> Beyond the summarization task shown in Figure 3, we agree with the reviewer’s point that the acceleration brought by MonarchAttention would be most obvious in very long sequence problems. Due to time and resource constraints, we cannot include experimental results in such settings for the rebuttal, but we plan to conduct experiments on a task from the Open Graph Benchmark to be included in the camera ready version.
>
> For future work, we also hope to team with researchers working on important long-context applications (who may hopefully also have sufficient computational resources) in order to test MonarchAttention on very large scale problems.
>
> *MonarchAttention only achieves improvements over vanilla attention if the number of optimization steps is in o(sqrt(T)). Although in the experimental evaluations T only reaches the modest size of 3, the paper lacks insights on the convergence rate of the optimization problem as a function of T.*
>
> We empirically find that alternating maximization usually converges in a couple of steps – to quantify this, we can measure the softmax variational objective f(M) in equation (6) after each step of alternating maximization in MonarchAttention for a particular ViT layer:
>
> | T     | 0    | 1    | 2    | 3    |
> |-------|------|------|------|------|
> | f(M)  | 418  | 937  | 940  | 940  |
>
>
> With T=2 we have already reached a stationary point in the objective. We will include these results in the camera ready version to emphasize the rapid convergence speed.
>
> *For well-definedness of the entropy terms, L and R must be constrained to be positive. I don’t see this mentioned anywhere in Section 3 before subsection “Alternating Maximization with Constraints”.*
>
> The reviewer is correct that we constrain L and R to be positive for the entropy term. We intended to delay discussion of this to the “Alternating Maximization with Constraints” subsection to solely focus the first half of Section 3 on computing the objective in equation (6). In the camera ready version, we will explicitly point out that we do not worry about the constraint on L, R until later in the section.
>
> *The authors state: “When L is fixed, the objective is concave in R”. This seems to be a typo, since the negative entropy is convex.*
>
> Since we are maximizing entropy (not negative entropy) in equation (6), the objective is indeed concave.
>
> *The chosen T is not always indicated in the experimental section, which initially gives the impression of it being quite large. I would suggest to explicitly state this for every setup to assure the reader no information is being hidden.*
>
> We thank the reviewer for this valuable suggestion – for the camera ready version, we will emphasize that T is very small throughout the paper, and provide explicit values in the main paper rather than the appendix.
>
> *A discussion of how Monarch attention integrates with GQA and MLA could be a noteworthy addition.*
>
> We believe that MonarchAttention should integrate with GQA/MLA quite seamlessly. From the modeling side, MonarchAttention is largely orthogonal to GQA/MLA, since they operate on the KV pairs prior to the attention operation. On the other hand, since the intermediate steps of MonarchAttention are FlashAttention-like calls, any optimizations of FlashAttention afforded by GQA/MLA would directly transfer to MonarchAttention.

---

> > ### Comment · Reviewer_nTcP · 2025-08-01
> >
> > I appreciate the efforts to include tasks from Open Graph Benchmark in the camera ready version.
> >
> > Thank you for providing a quantitative analysis of the convergence of alternating maximization. I still believe that theoretical convergence rates would additionally strengthen the paper, but agree that with the newly provided results empirical soundness is established.
> >
> > Concerning the concavity/convexity discussion, I agree with your assessment and believe that I was led astray by a different typo. The right hand side expansion of $f(B_{jk}; \tilde Q_j, K_k)$ has two sign errors, i.e., it expands to negative entropy rather than entropy. It would be great to correct this in the camera ready version.
> >
> > Thank you for otherwise addressing my comments and reaffirming the strength of the paper.

---

### Official Review · Reviewer_jzk9 · 2025-07-01

**Clarity:** 4
**Significance:** 4
**Originality:** 4
**Rating:** 6
**Confidence:** 3

**Summary:**

Attention requires quadratic complexity, sub-quadratic approximation via Monarch matrices (structured) may improve complexity. Transferable, with no additional training and hardware-efficient make it a potentially good way of reducing attention overhead.

**Questions:**

I think this is a good paper, and it just ‘works’, which is great. However, I would really love to see the ‘best possible’ sub-quadratic variant of _any one_ of these models (even monarch-attention!). This is optional, but can help. I am very option to raise my rating to an accept once I hear the author's views on this.

**Ethical Concerns:**

["NO or VERY MINOR ethics concerns only"]

**Final Justification:**

I agree with the rebuttal by authors, and think this paper should definitely be accepted + highlighted.

**Quality:**

4

**Strengths And Weaknesses:**

Strengths:
- Flashattn style kernels on quadratic-size blocks is very useful and can have big impact on performance.
- Zero-shot implies faster adoption.
- Very well written paper, easy to follow.
Weaknesses:
- Is it possible to have Figure 2, but with train-from-scratch variants as well on performer, cosformer, linear attention etc? Even a single variant (just to upper bound sub-quadratic attention) would be very insightful, not to serve as a ‘negative’ signal but to ~upper bound the performance of sub-quadratic attention.

---

> ### Author Rebuttal · Authors · 2025-07-29
>
> We thank the reviewer for the positive reception.
>
> The suggestion to upper bound performance of sub-quadratic attention is very interesting, but unfortunately training any of the models considered in the paper from scratch is currently infeasible due to time and resource constraints. As an alternative, we repeat the BART summarization experiment but fine-tuning *with* Nystromformer and MonarchAttention (instead of zero-shot replacement), which we believe is a reasonable proxy for the “best possible” sub-quadratic model:
>
> |     Softmax     | MA (fine-tuned) | MA (zero-shot) | Nyst. (fine-tuned) | Nyst. (zero-shot) |
> |-----------------|-----------------|----------------|--------------------|-------------------|
> |      35.57      |      35.38      |      33.94     |       34.60        |       32.86       |
>
> According to this, fine-tuned MonarchAttention is very close to the original softmax model and may act as an upper bound on sub-quadratic attention performance. We will include these results in the camera ready version. Overall, this paper focuses on zero-shot approximation, but we believe that MonarchAttention is a strong alternative to attention when training from scratch, to be explored in future work.

---

> ### Comment · Reviewer_jzk9 · 2025-08-05
>
> Thank you for addressing my concerns, I am happy with the paper and will raise my score further to a strong accept.

---

### Official Review · Reviewer_DpNT · 2025-07-03

**Clarity:** 3
**Significance:** 3
**Originality:** 2
**Rating:** 4
**Confidence:** 3

**Summary:**

This paper proposes MonarchAttention, which approximates each attention layer by fitting a structured Monarch matrix, enabling sub-quadratic attention without any model retraining. In zero-shot swaps on ViT, RoBERTa, BART, and DiT-XL, it cuts attention FLOPs by a large margin while incurring minor accuracy drops.

**Questions:**

**Questions**:

1. Per-Layer Error Analysis? The paper reports only end-to-end task metrics when swapping *all* layers or a fixed prefix. Some layers (e.g., early vs. late) may tolerate approximation much better. I was wondering if a layer-based ablation might offer more insights into where MonarchAttention is safe and where it breaks.
2. How do you select the block dimensions m and b for arbitrary sequence lengths? Have you evaluated the impact of padding or uneven block sizes on approximation quality at sequence ends?

**Ethical Concerns:**

["NO or VERY MINOR ethics concerns only"]

**Final Justification:**

Post-rebuttal final justification: the reviewer thanks the author for the in-depth rebuttal and addressing some of the reviewer's concerns. I will maintain the original score.

**Limitations:**

yes

**Quality:**

3

**Strengths And Weaknesses:**

**Strengths**:

The paper tests MonarchAttention on pretrained models without any fine-tuning or weight updates and achieves sub-quadratic complexity. Experiments are done across various model architectures and benchmarks.

**Weaknesses**:

1. The paper doesn’t show any experiments showing that MonarchAttention remains stable after fine-tuning, which might be critical for any downstream applications. I wonder if the authors have attempted any post-swap fine-tuning with MonarchAttention in place, and if so, how stable is training compared to the original softmax?
2. The same rank-one block approximation is applied uniformly to every row of the attention matrix. In some tasks, however, some tokens require more fine-grained attention while others need only broad context. The rigid Monarch factorization cannot adaptively allocate more capacity to certain rows, potentially harming accuracy on critical positions.

---

> ### Author Rebuttal · Authors · 2025-07-29
>
> We thank the reviewer for the positive feedback and thoughtful remarks. We address each of the reviewer’s comments and questions in the following.
>
> *The paper doesn’t show any experiments showing that MonarchAttention remains stable after fine-tuning, which might be critical for any downstream applications. I wonder if the authors have attempted any post-swap fine-tuning with MonarchAttention in place, and if so, how stable is training compared to the original softmax?*
>
> Although our paper highlights the zero-shot capabilities of MonarchAttention, the reviewer brings up an interesting point regarding training with MonarchAttention. We repeat the fine-tuning procedure for obtaining the summarization model with N=8192 reported in Figure 3, except we replace the softmax attention layers with MonarchAttention prior to fine-tuning. We show the ROUGE-1 results compared with zero-shot replacement:
>
> |     Softmax     | MA (fine-tuned) | MA (zero-shot) | Nyst. (fine-tuned) | Nyst. (zero-shot) |
> |-----------------|-----------------|----------------|--------------------|-------------------|
> |      35.57      |      35.38      |      33.94     |       34.60        |       32.86       |
>
> In summary, training with MonarchAttention is stable and in fact improves on zero-shot substantially. We will include these results in the camera ready version.
>
> *The same rank-one block approximation is applied uniformly to every row of the attention matrix. In some tasks, however, some tokens require more fine-grained attention while others need only broad context. The rigid Monarch factorization cannot adaptively allocate more capacity to certain rows, potentially harming accuracy on critical positions.*
>
> We certainly agree that some tokens require more degrees of freedom compared to others, and we believe that our approach can be generalized to structured matrices with variably-sized rank-one blocks – we will include a discussion of this in the camera ready version. However, an adaptive method that identifies important tokens would likely introduce additional overhead (as with LSH-based approaches), whereas our method is a generic yet fast approximation that works well in practice. We note that Monarch is an expressive class of matrices that can already represent many interesting structured matrices (Proposition 3.2 in [1]).
>
> [1] Dao et al. Monarch: Expressive Structured Matrices for Efficient and Accurate Training.
>
> *Per-Layer Error Analysis? The paper reports only end-to-end task metrics when swapping all layers or a fixed prefix. Some layers (e.g., early vs. late) may tolerate approximation much better. I was wondering if a layer-based ablation might offer more insights into where MonarchAttention is safe and where it breaks.*
>
> We appreciate the reviewer’s suggestion on this – in addition to the results shown in Table 1, we conduct some more experiments here to gauge the sensitivity to choice of layer for replacing attention. For the RoBERTa SQuAD experiment, we consider replacing different layers with MonarchAttention (keeping the rest intact):
>
> | Layer | 0    | 1    | 2    | 3    | 4    | 5    | 6    | 7    | 8    | 9    | 10   | 11   |
> |-------|------|------|------|------|------|------|------|------|------|------|------|------|
> | F1    | 92.7 | 92.5 | 92.6 | 92.0 | 91.4 | 91.9 | 86.3 | 91.1 | 92.7 | 93.2 | 93.0 | 92.3 |
>
>
> According to this, the initial and final layers are more amenable to replacement compared to the middle layers for BART. On the other hand, we also provide a more fine-grained version of Table 1 by replacing quarters (rather than halves) of the DiT model, where replacing earlier layers is more favorable:
>
> | Quarter |  FID   |  sFID  |  FLOPs (10^9) |
> |---------|--------|--------|---------------|
> | First   |  0.07  |  0.14  |      7.20     |
> | Second  |  0.21  |  0.39  |      7.20     |
> | Third   |  0.34  |  0.52  |      7.20     |
> | Fourth  |  1.35  |  2.23  |      7.20     |
>
> We will include these additional experiments in the camera ready version.
>
> *How do you select the block dimensions m and b for arbitrary sequence lengths? Have you evaluated the impact of padding or uneven block sizes on approximation quality at sequence ends?*
>
> First, we note that varying block size b yields a tradeoff between FLOPs and model quality (b=1 or N gives back dense representation, m=b=sqrt(N) is least compute) – this is demonstrated in the RoBERTa SQuAD experiment shown in the right panel of Figure 2. In this sense, we simply choose b (m is just the ceiling of N/b) according to our computational budget. In the case of ViT with N=197=14*14+1, there is an obvious choice of b=14 for the number of patches along the width of the image, and pre-padding since the cls token is at the beginning of the sequence – in fact, if we use post-padding as opposed to pre-padding, accuracy drops from 93.26% to 92.59% for T=2. We will include discussion for these hyperparameter choices in the camera ready version.

---

### Note · Authors · 2025-08-12

Dear Area Chair,

We thank all the reviewers for their time and valuable feedback. Although the assessment has been broadly positive, we believe our rebuttal has successfully addressed the main points raised, particularly regarding flexibility and robustness of our approach. We are confident in the contributions of our work and hope you will consider it favorably for acceptance.

Sincerely, The Authors

---

### Decision · Program_Chairs · 2025-09-17

**Decision:**

Accept (spotlight)

**Comment:**

**Paper summary**

To speed up the inference time for transformer models authors propose to approximate attention (quadratic in sequence length) with Monarch matrices introducing MonarchAttention which is sub-quadratic in sequence length. Authors propose this approximation in a zero-shot manner: model is used as is without any fine-tuning and approximation is done via variational form of softmax within 2-3 optimization steps (empirical results) thus preserving sub-quadratic complexity. The paper also provides and discusses the hardware-efficient with the proper memory/IO implementation. As a result authors get 1.4x speed up compared to FlashAttention2.0, better approximation than linear attention for several tasks (4 different models / tasks in vision and language were considered).

**Arguments taken for decision making**

*Strengths*
- **Reviewers jzk9, nTcP** pointed out that the paper is clearly and well written, easy to follow
- **All reviewers** pointed out on a wide applicability and potential (e.g. graph transformers and large reasoning models) as the method is zero-shot without any fine-tuning and reduces compute from quadratic to sub-quadratic complexity
- **Reviewer DpNT, nTcP** pointed out extensive empirical results (including across various model architectures and benchmarks)

*Weaknesses*
- Resolved
	- **Reviewer jzk9** pointed out absence of upper bound performance of sub-quadratic attention. During rebuttal due to time constraints authors provided a variant of upper bound by fine-tuning a model with MonarchAttention after softmax zero-shot approximation and the reviewer was satisfied.
    	- I also agree with the authors points that work primarily focused on zero-shot, so this is not a critical issue for paper acceptance in my view.
	- **Reviewer nTcP** pointed out that the paper lacks insights on the convergence rate of the optimization problem for softmax approximation and this could be in the end $sqrt(N)$ steps making it again quadratic. During rebuttal authors provided empirical justification showing that in 2-3 steps only optimization saturates and reaches optimal point. The reviewer is satisfied with empirical support though theoretical guarantees will be great to have.
	- **Reviewer DpNT** asked for per layer analysis and some clarifications which authors provided during rebuttal and simply can be added into appendix.
	- **Reviewer DpNT** raised issue that some positions may need fine-grained attention and approximation can hurt. After rebuttal and discussion the reviewer agreed that there are trade offs between quality and practicality.
- Partially resolved / not resolved
	- **Reviewer nTcP** pointed out on absence of empirical results in the most impactful areas e.g., graph attention and very long-context settings. During rebuttal authors mentioned that they are working on Open Graph Benchmark and will provide results in camera ready due to time constraints. Authors also mentioned that ideal test will be very long sequence problems.
    	- I agree that these tasks could strengthen the paper even more  and speed up adoption, but authors listed them as limitations in the paper, and I believe results presented are strong to encourage the community to try broader experiments with MonarchAttention.
	- **Reviewer DpNT** pointed out that while the method works in zero-shot, it is unclear if it will be stable in fine-tuning regime as for downstream tasks we often need to fine-tune models. During rebuttal authors provided results of first approximating with MonarchAttention and then fine-tuning that model on summarization task showing that it is stable and given close to softmax attention results. However the reviewer agreed only partially with results as the training had the same loss / data as pre-training.
    	- I think the solution could be to just fine-tune the model with softmax attention for any new downstream task and new objective and after that do zero-shot approximation with MonarchAttention. For this angle as zero-shot inference speeding up, I think, the paper has strong empirical justification. Anything related to training with MonarchAttention speeding up the training itself, I believe, is worth separate paper and was out of authors' focus.

**Recommendation summary**

All reviewers (with different degrees) support the paper acceptance after discussion acknowledging novelty, well written text, extensive empirical justification and potential high impact of the work. Reviewers also asked to extend results to training from scratch and fine-tuning for downstream tasks: overall authors listed these as limitations and I agree that the work potentially can grow into bigger applications widespreading across different domains and tasks, pre-training and post-training stages, and all pieces hard to cover and include in one paper. Current results authors provided I think are enough per NeurIPS standards showing applicability and speed up for inference only and I am looking forward to seeing bigger impacts on the community opening space for a variety of works to make the paradigm shift (applying MonarchAttention to speed up training). Given all this, **I recommend strong acceptance of the paper.**

**Recommendation to authors**

Dear authors, please include important pieces and clarifications from the discussion into the final version of the paper as you promised. If possible, it will be great to have Open Graph Benchmark added as requested by Reviewer nTcP (this can speed up adoption), proper fine-tuning results as requested by Reviewer DpNT (this can speed up adoption), and proper upper bound as requested by Reviewer jzk9 (this can motivate a lot for the work extension showing potential even for training from scratch). But the later is optional to me as your current results already show high potential impact for inference and leaving training applicability as a future work.